# Increased Risk of Hospitalization for Pneumonia in Italian Adults from 2010 to 2019: Scientific Evidence for a Call to Action

**DOI:** 10.3390/vaccines11010187

**Published:** 2023-01-16

**Authors:** Emanuele Amodio, Francesco Vitale, Daniela d’Angela, Ciro Carrieri, Barbara Polistena, Federico Spandonaro, Alessandra Pagliaro, Eva Agostina Montuori

**Affiliations:** 1Department of Health Promotion, Mother and Child Care, Internal Medicine and Medical Specialties, University of Palermo, Piazza delle Cliniche 1, 90127 Palermo, Italy; 2C.R.E.A. Sanità (Centre for Applied Economic Research in Healthcare), 00196 Rome, Italy; 3Department of Economic and Finance, University of Rome Tor Vergata, 00133 Rome, Italy; 4San Raffaele University, 20132 Milan, Italy; 5Medical Department, Pfizer, 00188 Rome, Italy

**Keywords:** pneumonia, hospitalizations, Italy, mortality

## Abstract

Background: Understanding trends in pneumonia-associated hospitalizations can help to quantify the burden of disease and identify risk conditions and at-risk populations. This study evaluated characteristics of hospitalizations due to pneumonia that occurred in Italy in a 10-year period from 2010 to 2019. Methods: All hospitalizations with a principal or secondary diagnosis of pneumonia over the 10-year period were included, which were identified by hospital discharges for all-cause pneumonia and pneumococcal pneumonia in the anonymized hospital discharge database of the Italian Health Ministry. Results: A total of 2,481,213 patients were hospitalized for pneumonia between 2010 and 2019; patients aged 75–86 years accounted for 30.1% of hospitalizations. Most hospitalizations (88.1%) had an unspecified pneumonia discharge code. In-hospital death was recorded in 13.0% of cases. The cumulative cost for pneumonia hospitalizations of the 10-year period were EUR 11,303,461,591. Over the observation period, the incidence rate for hospitalized all-cause pneumonia in any ages increased from 100 per 100,000 in 2010 to over 160 cases per 100,000 per year in 2019 (*p* < 0.001). Overall, there was a significant increase in annual percent changes in hospitalization rates (+3.47 per year), in-hospital death (+4.6% per year), and costs (+3.95% per year) over the 10-year period. Conclusions: Our analysis suggests that hospitalizations for pneumonia are increasing over time in almost all age groups, especially in the elderly. Given the substantial burden of pneumonia in terms of mortality, healthcare resources, and economic costs, greater public health efforts should thus be made to promote vaccinations against influenza and pneumococcus, particularly in high-risk groups.

## 1. Introduction

Community-acquired pneumonia (CAP) is defined as an acute infection of the lung that is acquired outside of a hospital setting [1]. Streptococcus pneumoniae and respiratory viruses are the most frequently identified pathogens in patients with CAP [1]. CAP is a leading cause of morbidity and mortality, especially in the elderly and in patients who are immunocompromised [1]. In addition to age, various chronic comorbidities such as heart disease, pulmonary disorders, and diabetes can increase the risk of pneumonia [2,3]. In a study from Spain, for example, the risk of hospitalization due to pneumococcal disease in patients with CAP and an underlying cardiac, respiratory, or metabolic comorbidity was 73-fold higher than that in patients with no comorbidity [4]. In the US, the observed annual age-adjusted incidence of patients hospitalized with CAP is 649 cases per 100,000 adults, corresponding to more than 15 million hospitalizations annually, with a mortality rate of 6.5% [5].

The etiology of CAP is often difficult to determine, due to difficulties in collecting representative respiratory specimens and inaccurate methods for microbial detection. Even with extensive testing, the cause of pneumonia can be identified in roughly one-third of patients [6]. Treatment of CAP usually is empirically later adapted according to the identified causative pathogen [1,7]. 

The incidence of pneumonia hospitalizations has been steadily increasing in recent years [8]. Understanding the burden and trends of community-pneumonia-associated hospitalizations can thus help to provide useful information to quantify the burden of the disease and identify at high-risk populations. Herein, we report incidence rates for CAP hospitalizations in adults in Italy between 2010 and 2019 and describe the demographics, underlying risk factors, and standard-of care etiological diagnosis among hospitalized pneumonia patients.

## 2. Materials and Methods

### 2.1. Study Design

This is a retrospective cohort study of the anonymized hospital discharge database of the Italian Health Ministry, summarized as part of a report from Crea Sanità, Rome, Italy, an independent center that carries out economic healthcare evaluations. Data of annual pneumonia hospitalizations in adults aged ≥ 18 years of age/in people of all ages for a 10-year period between 2010 and 2019 were extracted. All-cause pneumonia as the key outcome of interest was defined as ICD 9 CM 480–486 in a principal or secondary diagnosis position pneumonia (Appendix A). Case definitions for other pneumonia definitions are defined in Appendix A and case definitions for underlying comorbidities are described in Appendix A. In addition, the following variables were collected: patient’s place of residency, date of hospital admission, age, sex, length of stay (in days), disposition on admission, discharge disposition, diagnosis related groups (drg), drg weight (measured as the ratio of the mean episode cost in a drg group to the mean cost of episodes in all drgs), and hospitalization costs (in euro). 

### 2.2. Statistical Analysis

Qualitative data are summarized as absolute frequencies and incidence rates (IR; cases per 100,000 population at risk per year). Denominators for incidence rates were calculated using the census population in Italy from 2010 to 2019. Quantitative data are reported as the mean (standard deviation, SD) if normally distributed, otherwise as a median (interquartile range, IQR). Join point analysis was used to evaluate trends over time of hospitalization rates, and average percentage change (APC) and average annual percent change (AAPC) were calculated. According to several studies, the join point analysis is often used to summarize trends in disease and mortality rates, and it is a common estimator that uses a linear model on the log of the age-standardized rates [9]. 

A *p*-value < 0.05 was considered statistically significant. Analyses were performed using R Software analysis (version 4.0.5, R Foundation for Statistical Computing, Vienna, Austria) and, as for joinpoint analysis, the packages “Segmented” and “Strucchange” were used (RStudio Team. RStudio: Integrated Development for R. RStudio; PBC: Boston, MA, USA, 2020). 

## 3. Results 

### 3.1. General Characteristics of Hospitalizations for Pneumonia

During the period from 2010 to 2019, there were a total of 2,481,213 hospital discharges for pneumonia (Table 1), of which 89.3% in the adult population (2,215,864 hospitalizations, IR = 440.3 per 100,000 per year). Somewhat more hospitalizations were seen among men (55.3%) than in women (44.7%), for an IR of 468.5 per 100,000 in males and 357.0 per 100,000 in females. Patients from 75–86 years (IR: 1688.5 per 100,000 per year) accounted for 30.1% of all hospitalizations for pneumonia, and the age group from 18–45 years (IR: 71.2 per 100,000 per year) had the lowest number of hospitalizations (6.1%). A slight increase in the number of hospitalizations with pneumonia as the main diagnosis is apparent starting around 2014. 

Absolute and relative frequencies of specified bacterial, viral and fungal causes are shown in Table 2. Pneumococcal pneumonia was responsible for about 20% of cases, followed by pneumonia from other specified bacteria (17%). Mycoplasma was responsible for almost 13% of cases and pseudomonas in 12%. A multitude of other micro-organisms and viruses were responsible for the remaining cases in smaller proportions.

### 3.2. Clinical Characteristics of Patients Hospitalized for Pneumonia

Pneumonia was recorded in the primary diagnostic field in 55.4% of cases and in any secondary diagnostic field in 44.6% of cases (Table 3). The vast number of hospitalizations (88.1%) had an unspecified diagnosis (unspecified pneumonia, 31.6%; unspecified bacterial pneumonia, 20.7%; unspecified bronchopneumonia, 35.8%). The median length of hospitalization was 12.3 days (IQR 12.1–12.5); median drg was 1.32 (IQR 1.27–1.36). Regarding comorbidities, 27.3% had at least one comorbidity; chronic respiratory disease (17.7%) was the most frequent, followed by cancer (11.1%) and diabetes (10.9%). While most patients (71.6%) had a routine discharge, of note, in-hospital death was recorded in 13.0% of cases (Table 4). Lastly, over the time period analyzed, average costs for hospitalization amounted to EUR 1,130,346.15 per year. 

### 3.3. Trends in Hospitalizations over Time

Trends in rates of hospitalizations for pneumonia according to clinical diagnosis are shown in Figure 1. While most of the different diagnoses for pneumonia appeared to be relatively stable during the analyzed time period, there was a marked and steady increase in hospitalizations for unspecified pneumonia from slightly more than 100 cases per 100,000 per year in 2010 to over 160 per 100,000 in 2019 as well as unspecified bacterial pneumonia (from about 65 per 100,000 to in 2010 to about 95 per 100,000 in 2019). With few exceptions, over 2010–2019 there was a significant increase in annual percent changes in hospitalization rates for pneumonia in Italy considering all variables assessed. Significant increases were seen in both men and women, in all age groups except for <18 years and 18–45 years, and in unspecified and specified diagnoses (except unspecified bronchopneumonia). In particular, hospitalization rates in subjects aged <18 years decreased significantly over time by 4.25% per year (from 341.2 per 100,000 in 2010 to 255.4 per 100,000 in 2019; *p* < 0.001). Moreover, significant increases were observed for all comorbidities as well as discharge diagnoses, including in-hospital death. Accordingly, costs for hospitalization and median length of stay also significantly increased by 3.95% and 0.31 (days per year), respectively.

## 4. Discussion

Pneumonia represents a serious public health issue that is associated with high medical burden and substantial economic costs, in addition to hospitalizations and mortality in individuals of all age groups [10]. Herein, we studied hospitalizations and in-hospital mortality in patients with main and secondary diagnoses of pneumonia in Italy over a 10-year period spanning 2010–2019. During this period, there were almost 2.5 million hospitalizations for pneumonia and more than 322,000 related in-hospital deaths. One of the main findings from our analysis arises that hospitalizations for unspecified pneumonia have steadily increased by about 60% during 2010–2019, with an in-hospital mortality rate of 13%. The increased number of hospitalizations for pneumonia was also apparent across all age groups and comorbidities assessed. In addition, during the 10-year period, total costs for hospitalization for pneumonia were well over EUR 11,000,000,000, confirming the huge economic burden of pneumonia for healthcare systems. In particular, considering that before the SARS-CoV-2 the total Italian health fund was about EUR 110 billion per year, we can estimate that hospitalizations for pneumonia in Italy every year requires more than 1% of total healthcare expenditure.

The annual incidence of hospitalizations for pneumonia observed herein are broadly similar to those reported in a US study assessing hospitalizations for CAP with an annual incidence of 649 patients per 100,000 adults using data from 2014–2016 [5]. In that analysis, mortality at 30 days was 13%, which compares well to the 13% observed herein (in-hospital deaths). In a Dutch study, in-hospital mortality from CAP in patients aged ≥65 years was 11.3% [11]. Similar to our data, the mean duration of hospitalization was 12.1 days [11] compared to a median of 12.3 in our study. As expected, the incidence of hospitalizations for pneumonia also increased with age as seen herein, with a rate of 69 per 100,000 in those 18–24 years and 3951 per 100,000 in those ≥85 years [5]. However, from our data it can be noted that the incidence rate among those <18 years was almost double that seen in the 18–45 year age group. Similar results were reported in a study from Canada, wherein hospitalizations for all-cause pneumonia were highest in children <5 years and in adults >70 years of age and rates of mortality (11.6–12.3%) similar to those observed in our study [12]. However, over time we found a statistically significant reduction in hospitalization rates in subjects aged <18 years, which could be in large part attributable to the increased pneumococcal vaccination coverage observed in children in Italy during the study period [13]. Our results are also relevant considering older that individuals who are hospitalized very often have comorbidities and complications that have a negative impact on health and recovery. In fact, admissions in patients ≥65 years of age represented 69.9% of all hospitalizations.

The increasing number of hospitalizations with an unspecified diagnosis of pneumonia from 2010–2019 is of further interest, with significant increases seen across all age subgroups and comorbidities. First, this indicates that bacterial testing is often not performed in-hospital, as previously reported [14]. Second, it highlights the need for preventive strategies as noted in recent studies, which stress the use of pneumococcal vaccination in older individuals and in those with risk factors such as chronic obstructive pulmonary disease, male gender, and diabetes [15,16]. Third, the lack of an etiologic diagnosis may increase the risk of overuse, inappropriate, or inefficient antimicrobial therapy and, ultimately, the selection of resistant and multi-drug resistant bacteria [17].

It is well known that pneumonia hospitalization trends in general population can be significantly modified by pneumococcal, influenza and COVID-19 vaccination coverage. In particular, influenza vaccination has been associated with reduced severity of CAP and improved overall survival during influenza seasons in patients with CAP [18]. In recent years, SARS-CoV-2 has also contributed to the pneumonia burden of diseases, although in our study this pathogen had no impact [19]. However, some studies have suggested that COVID-19 vaccines provide high protection against pneumonia and reduce severity of pneumonia [20,21].

Pneumococcal pneumonia was responsible for about one-fifth of all cases of hospitalizations for pneumonia in the present study. Vaccination against pneumococcus has been available since 1983, and immunization has substantially reduced the burden of pneumococcal disease in children and the elderly [22]. A study on the efficacy of a polysaccharide conjugate vaccine (PCV), PCV-13, against vaccine-type pneumococcal pneumonia has reported a vaccine efficacy of 46% among older adults and 40% among older adults with underlying at-risk conditions [23,24]. A vaccine effectiveness of around 70% has been reported for PCV-13 in older adults hospitalized for vaccine-type community-acquired pneumonia in the US [25]. These results are in contrast to a more recent analysis of PPV-23 which suggested that the vaccine does not prevent hospitalizations due to community-acquired pneumonia [26]. However, it would seem intuitive that the introduction of vaccines covering additional subtypes would be advantageous [27]. In most countries, the polysaccharide PCV-13 or PPV-23 vaccines (or both) are currently recommended in older individuals [1].

Notwithstanding, while rates of pneumococcal disease caused by specific serotypes of *Streptococcus pneumoniae* may decrease substantially with vaccination, pneumonia caused by other serotypes may increase proportionally and thus continuous monitoring should be implemented in order to evaluate the epidemiological impact of pneumococcal disease and vaccination strategies. In a study in England, for example, the rate of pneumococcal disease caused by PCV-7 types decreased by 97% after the introduction of the PCV-7 vaccine, although the rates of pneumonia caused by non-PCV-7 serotypes later doubled [28]. In a recent analysis in the US, it was noted that compared to PCV13, the additional serotypes covered by PCV-15 and PCV-20 still have a substantial contribution to the clinical and economic burden of pneumococcal disease [29]. The newly licensed PCV-20 and PCV-15 help to address the unmet need of protecting adults against the increased burden of pneumococcal disease due to serotypes not included in PCV13. The long-term benefits of vaccination with next-generation vaccines will depend on the evolution of pneumococcal serotypes over time, which is difficult to predict.

There are limitations to our study. Hospital claims data are subject to outcome misclassification and bias, which may have impacted the validity of our findings. This aspect depends on the quality of the codifying report, and we cannot exclude variation of the codification process occurred between different regions or in different years. Additionally, our analysis has only included hospitalized pneumonia cases and no conclusions can be drawn on the overall CAP burden in Italy. In this sense, it should be also noted that the difficulty in microbial detection and the lack of information about pneumococcal vaccination status in hospitalized patients may also affect the true trends of pneumococcal pneumonia as well as the evaluation of vaccine effectiveness.

Despite these limitations, our study shows that hospitalizations for pneumonia have increased between 2010 and 2019, especially in the elderly and frail subjects. Unfortunately, there is no data on the current coverage of pneumococcal vaccination in subjects ≥65 years of age. Our data highlight that pneumonia is responsible for the substantial burden in terms of mortality, healthcare resources, and economic costs. Greater public health efforts should thus be made to promote vaccination against respiratory infectious diseases, especially in subjects at a higher risk for severe sequelae.

## Figures and Tables

**Figure 1 vaccines-11-00187-f001:**
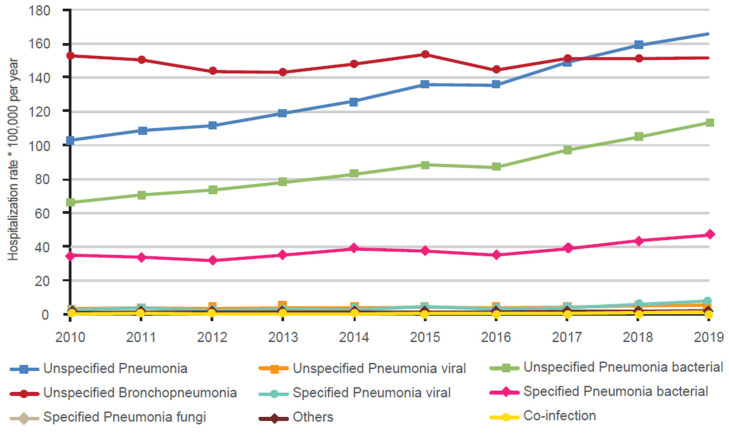
Trends in hospitalization rates over time according to clinical diagnosis in Italy (2010–2019).

**Table 1 vaccines-11-00187-t001:** General characteristics of hospitalizations for pneumonia in Italy from 2010 to 2019.

		Total Number of Pneumonia Hospitalizations	Incidence Rate Per 100,000	%
**Total**		2,481,213	411.2	100%
Gender				
	Male	1,372,040	468.5	55.3%
	Female	1,109,103	357	44.7%
Age (years)				
	<18	265,354	264.9	10.7%
	18–45	150,804	71.2	6.1%
	46–65	330,032	196.1	13.3%
	66–75	411,426	651.1	16.6%
	76–85	747,507	1688.50	30.1%
	>85	576,095	3681.60	23.2%
Year				
	2010	218,152	365.4	8.8%
	2011	223,389	374.1	9.0%
	2012	220,349	369.0	8.9%
	2013	229,607	384.6	9.3%
	2014	241,632	404.7	9.7%
	2015	255,607	428.1	10.3%
	2016	246,460	412.8	9.9%
	2017	267,217	447.5	10.8%
	2018	282,850	473.7	11.4%
	2019	295,950	495.7	11.9%

**Table 2 vaccines-11-00187-t002:** Absolute and relative frequencies of specified bacterial, viral and fungal causes.

ICD9-CM Diagnosis Code	ICD9-CM Diagnosis Description	Diagnostic Groups	Total Specified Cases 2010–2019	% Specified Cases out of Total Specified Cases 2010–2019
481	Pneumococcal pneumonia (Streptococcus pneumoniae)	Specified Pneumonia bacterial	53,879	20.44%
482.89	Pneumonia from other specified bacteria	Specified Pneumonia bacterial	44,947	17.05%
483.0	Mycoplasma Pneumoniae pneumonia	Specified Pneumonia bacterial	33,532	12.72%
482.1	Pseudomonas pneumonia	Specified Pneumonia bacterial	32,106	12.18%
482.41	Staphylococcus aureus pneumonia	Specified Pneumonia bacterial	18,239	6.92%
482.0	Klebsiella pneumoniae pneumonia	Specified Pneumonia bacterial	15,824	6.00%
487.0	Influenza with pneumonia	Specified Pneumonia viral	14,779	5.61%
482.82	Escherichia coli pneumonia [E. coli]	Specified Pneumonia bacterial	7226	2.74%
484.6	Pneumonia in aspergillosis	Specified Pneumonia fungi	6525	2.47%
483.1	Chlamydia Pneumonia	Specified Pneumonia bacterial	6061	2.30%
482.2	Haemophilus influenzae pneumonia (H. influenzae)	Specified Pneumonia bacterial	5777	2.19%
482.30	Streptococcus pneumonia, unspecified	Specified Pneumonia bacterial	5301	2.01%
480.1	Respiratory syncytial virus pneumonia	Specified Pneumonia viral	4914	1.86%
482.40	Staph pneumonia, unspecified	Specified Pneumonia bacterial	3094	1.17%
482.39	Pneumonia from other Streptococci	Specified Pneumonia bacterial	2597	0.99%
484.1	Cytomegalovirus pneumonia	Specified Pneumonia viral	2548	0.97%
480.0	Adenovirus Pneumonia	Specified Pneumonia viral	2359	0.89%
480.2	Parainfluenza virus pneumonia	Specified Pneumonia viral	2154	0.82%
482.31	Streptococcus pneumonia, group A	Specified Pneumonia bacterial	965	0.37%
482.32	Streptococcus pneumonia, group B	Specified Pneumonia bacterial	390	0.15%
484.3	Pneumonia in whooping cough	Specified Pneumonia bacterial	225	0.09%
480.3	SARS pneumonia—Associated coronavirus	Specified Pneumonia viral	163	0.06%
484.5	Pneumonia in anthrax	Specified Pneumonia bacterial	20	0.01%
482.4	Staph pneumonia	Specified Pneumonia bacterial	14	0.01%

**Table 3 vaccines-11-00187-t003:** Clinical and administrative characteristics of hospitalizations for pneumonia in Italy from 2010 to 2019.

		Total Number of Pneumonia Hospitalizations	Incidence Rate Per 100,000	% *
ICD-9 Pneumonia Code Position				
	Principal position	1,374,227	227.7	55.4%
	Second position	1,106,991	183.4	44.6%
ICD-19 Pneumonia Code				
	Unspecified pneumonia	783,574	129.8	31.6%
	Unspecified pneumonia viral	25,290	4.2	1.0%
	Unspecified pneumonia bacterial	514,174	85.2	20.7%
	Unspecified bronchopneumonia	889,320	147.4	35.8%
	Specified pneumonia viral	24,265	4.00	1.0%
	Specified pneumonia bacterial	224,320	37.20	9.0%
	Specified pneumonia fungal	5719	0.90	0.2%
	Other	10,110	1.70	0.4%
	Coinfection	4441	0.70	0.2%
Comorbidities				
	Chronic respiratory disease	438,730	72.7	17.7%
	Chronic cardiovascular disease	173,437	28.7	7.0%
	Chronic kidney disease	170,136	28.2	6.9%
	Cancer	274,394	45.5	11.1%
	Diabetes mellitus	271,262	45	10.9%
	At least one comorbidity	677,903	112.3	27.3%
	At least two comorbidities	230,401	38.2	9.3%
	Three or more comorbidities	52,023	8.6	2.1%
Discharge disposition				
	Routine discharge	1,775,817	294.3	71.6%
	Transfer	191,474	31.7	7.7%
	In-hospital death	322,474	53.4	13.0%
	Other	191,453	31.7	7.7%

* column percentages.

**Table 4 vaccines-11-00187-t004:** Annual percent changes in hospitalization rates for pneumonia in Italy from 2010 to 2019 stratified by different variables.

Variable		Annual Percent Change	*p*-Value
Raw hospitalization rates		+3.47	<0.001
Gender	Male	+3.08	<0.001
	Female	+3.93	<0.001
Age (years)	<18	−4.25	0.004
	18–45	−1.02	0.304
	46–65	+3.67	<0.001
	66–75	+3.53	<0.001
	76–85	+4.32	0.003
	>85	+7.49	0.003
Diagnosis (Position)	Pneumonia in first diagnosis	+1.35	0.011
	Pneumonia in secondary diagnosis	+6.18	0.003
Diagnosis	Unspecified pneumonia	+5.48	0.002
	Unspecified pneumonia viral	+4.60	0.009
	Unspecified pneumonia bacterial	+5.86	<0.001
	Unspecified bronchopneumonia	+0.10	0.761
	Specified pneumonia viral	+11.62	0.003
	Specified pneumonia bacterial	+3.33	0.003
	Specified pneumonia fungi	+6.38	0.001
	Other	+5.99	<0.001
	Coinfection	+16.11	0.001
Comorbidities	Chronic respiratory diseases	+3.60	<0.001
	Chronic cardiovascular diseases	+2.85	<0.001
	Chronic kidney diseases	+5.21	<0.001
	Cancer	+3.02	<0.001
	Diabetes	+1.86	<0.001
Comorbidities (number)	At least one comorbidity	+3.76	<0.001
	At least two comorbidities	+2.90	<0.001
	Three or more comorbidities	+2.19	0.002
Discharge disposition	Routine discharge	+2.13	0.001
	Transfer	+9.08	<0.001
	In-hospital death	+4.56	<0.001
	Other	+9.00	<0.001
Hospitalization costs (euro)		+3.95	<0.001
In-hospital death		+4.60	<0.001
Length of stay (days)		+0.31	0.038
DRG weight *		+0.57	0.003

* Measured as the ratio of the mean episode cost in a drg group to the mean cost of episodes in all drgs.

## Data Availability

Data analysed in this study will be made available, as aggregates, following reasonable request sent to the corresponding Author of the article.

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
