# Peer review of "Increased Risk of Hospitalization for Pneumonia in Italian Adults from 2010 to 2019: Scientific Evidence for a Call to Action"

_vaccines, 2023, doi:10.3390/vaccines11010187_

Round 1
Reviewer 1 Report
This is a brief report that examined the trends in pneumonia-associated hospitalizations in Italy over a 10-year period obtained from the discharge database of the Italian Health Ministry. The main findings were that hospitalizations for pneumonia appeared to be increasing, especially in the elderly, with a significant burden in terms of mortality, and utilization of healthcare resources and costs. These findings suggest that greater public health efforts need to be made to promote vaccinations, such as influenza and pneumococcal vaccinations, especially in high-risk groups. The manuscript is well written and provides important information.
Comments
1) One concern, as addressed party by the authors in the limitations of the study, relates to whether these cases have been correctly coded. However, by the way they are coded, can the conclusion be reached that these are all community-acquired pneumonia cases?
2) Although most cases appear to be of unspecified pneumonia, there are some of specified bacterial, viral and fungal causes. Are details of this known, since it may be of interest if presented to the reader?
Author Response
Comment: This is a brief report that examined the trends in pneumonia-associated hospitalizations in Italy over a 10-year period obtained from the discharge database of the Italian Health Ministry. The main findings were that hospitalizations for pneumonia appeared to be increasing, especially in the elderly, with a significant burden in terms of mortality, and utilization of healthcare resources and costs. These findings suggest that greater public health efforts need to be made to promote vaccinations, such as influenza and pneumococcal vaccinations, especially in high-risk groups. The manuscript is well written and provides important information.
Answer: Dear Reviewer, we thank you for your endorsement to our findings that we hope could be of help to increase the scientific knowledge on this very important health topic. Below you will find a point-by-point answer to each raised question.
Q: One concern, as addressed party by the authors in the limitations of the study, relates to whether these cases have been correctly coded. However, by the way they are coded, can the conclusion be reached that these are all community-acquired pneumonia cases?
A: As you rightly observe, unfortunately, in our analysis we included all hospitalized pneumonia cases and, probably, at least a part of these pneumonia could be attributable to the hospitalization thus being hospital-acquired infections. Tring to quantify the real burden of the two infections (community and hospital acquired) could be at high risk of unprecision and, thus, we prefer to indicate this as a possible limit of the study (as reported in the Discussion section).
Q: Although most cases appear to be of unspecified pneumonia, there are some of specified bacterial, viral and fungal causes. Are details of this known, since it may be of interest if presented to the reader?
A: Thank you very much for this suggestion. In the revised version of the manuscript we included a supplementary table with absolute and relative frequencies of specified bacterial, viral and fungal causes.
Reviewer 2 Report
In the current study author's did a great job of Quantifying the illness burden, identifying risk factors, and identifying at-risk groups can all be aided by understanding patterns in pneumonia-related hospitalizations. This study analyzed the features of pneumonia hospitalizations that took place in Italy over a ten-year period, from 2010 to 2019.
According to our data, hospitalizations for pneumonia are on the rise across practically all age groups, but especially among the elderly. Greater public health efforts should therefore be undertaken to promote vaccinations against influenza and pneumococcus, particularly in high-risk populations, given the significant burden of pneumonia in terms of mortality, healthcare resources, and economic expenses.
My recommendation is to Accept in present form
Author Response
Comment: In the current study author's did a great job of Quantifying the illness burden, identifying risk factors, and identifying at-risk groups can all be aided by understanding patterns in pneumonia-related hospitalizations. This study analyzed the features of pneumonia hospitalizations that took place in Italy over a ten-year period, from 2010 to 2019. According to your data, hospitalizations for pneumonia are on the rise across practically all age groups, but especially among the elderly. Greater public health efforts should therefore be undertaken to promote vaccinations against influenza and pneumococcus, particularly in high-risk populations, given the significant burden of pneumonia in terms of mortality, healthcare resources, and economic expenses. My recommendation is to Accept in present form.
Answer: Dear Reviewer, we thank you for your appreciation and kind words for our study findings.